# Comparative Analysis of Cyclization Techniques in Stapled Peptides: Structural Insights into Protein–Protein Interactions in a SARS-CoV-2 Spike RBD/hACE2 Model System

**DOI:** 10.3390/ijms25010166

**Published:** 2023-12-21

**Authors:** Sára Ferková, Ulrike Froehlich, Marie-Édith Nepveu-Traversy, Alexandre Murza, Taha Azad, Michel Grandbois, Philippe Sarret, Pierre Lavigne, Pierre-Luc Boudreault

**Affiliations:** Department of Pharmacology and Physiology, Faculty of Medicine and Health Sciences, Institut de Pharmacologie de Sherbrooke, Université de Sherbrooke, 3001 12e Avenue Nord, Sherbrooke, QC J1H 5N4, Canada; sara.ferkova2@usherbrooke.ca (S.F.); marie-edith.nepveu-traversy@usherbrooke.ca (M.-É.N.-T.); alexandre.murza@usherbrooke.ca (A.M.); taha.azad@usherbrooke.ca (T.A.);

**Keywords:** peptidomimetics, protein–protein interaction (PPI), SARS-CoV-2, circular dichroism (CD), nuclear magnetic resonance (NMR)

## Abstract

Medicinal chemistry is constantly searching for new approaches to develop more effective and targeted therapeutic molecules. The design of peptidomimetics is a promising emerging strategy that is aimed at developing peptides that mimic or modulate the biological activity of proteins. Among these, stapled peptides stand out for their unique ability to stabilize highly frequent helical motifs, but they have failed to be systematically reported. Here, we exploit chemically diverse helix-inducing *i*, *i* + 4 constraints—lactam, hydrocarbon, triazole, double triazole and thioether—on two distinct short sequences derived from the N-terminal peptidase domain of hACE2 upon structural characterization and in silico alanine scan. Our overall objective was to provide a sequence-independent comparison of α-helix-inducing staples using circular dichroism (CD) and nuclear magnetic resonance (NMR) spectroscopy. We identified a 9-mer lactam stapled peptide derived from the hACE2 sequence (His34-Gln42) capable of reaching its maximal helicity of 55% with antiviral activity in bioreporter- and pseudovirus-based inhibition assays. To the best of our knowledge, this study is the first comprehensive investigation comparing several cyclization methods with the goal of generating stapled peptides and correlating their secondary structures with PPI inhibitions using a highly topical model system (i.e., the interaction of SARS-CoV-2 Spike RBD with hACE2).

## 1. Introduction

Protein–protein interactions (PPIs), which involve the complex, highly dynamic and complementary surfaces of two proteins, regulate a variety of intra- and extra-cellular biological processes. One example of such processes is the initial contact of pathogens with eucaryotic cells, such as SARS-CoV-2 [1,2,3]. Despite the promising therapeutic value of controlling PPIs via inhibition or activation, this topic has remained largely underexplored for several reasons. First, the PPI interface is composed of large polar contact regions ranging in surface area from 1500–3000 Å^2^; the interface often additionally includes shallow binding pockets that constitute 200–900 Å^2^ [4,5]. Such surfaces are not convenient for traditional small-molecule drugs (200–500 Da) [6,7]. Without a defined and confined pocket to target, it remains a challenge to design a molecule capable of interfering effectively in the PPI. Moreover, high-affinity interactions are ensured by either continuous and/or distinct “hot spot” regions and hydrophobic patches rich in Tyr, Trp, Leu, Ile, Phe and side chains of Arg or Lys [8]. Therefore, the high affinity of natural PPI interfaces requires potent inhibitors to effectively disrupt the majority of these heterogeneous interactions. Finally, screening for PPI modulators often requires specialized functional assays such as protein complementation assays or biophysical techniques. These methods might not perform as well as traditional high-throughput enzymatic assays. To overcome this dauting challenge—which is often referred to as an “undruggable” target [9]—researchers have largely relied on the development of macrocycles [10], neutralizing antibodies [11,12] and high-molecular-weight inhibitors based on the binding motifs of the complementary protein [13].

Peptidomimetics closely emulate the conformation of a natural protein, and they often retain its ability to interact with the biological target [14]. This class of compounds possesses distinct features that render them promising candidates for PPIs. For instance, they can effectively counteract the highly dynamic behavior of PPI involved proteins, and they can cover a large surface area (>800 Å^2^) with a high affinity and specificity. Given that 62% of PPI interfaces are made up of helical motifs [15] formed from 4–15 amino acids leading to 1–4 α-helical turns, α-helix-based mimetics represent promising tools.

Synthetic low-molecular-weight peptides that mimic bioactive α-helical conformations hold potential for drug design and optimization. Depending on the desired pharmacodynamic (PD) and pharmacokinetic (PK) parameters, two residues facing the same interface might be “stapled” using various cyclization techniques (Figure 1) to compensate for their lack of pharmacological properties (e.g., aqueous versus membrane solubility, metabolic stability, high renal clearance, etc.) [16]. The stapling technique is therefore viewed as a way of simultaneously optimizing both bioactive conformation and PK/PD properties. However, conformation alterations due to changes in the staple can have detrimental effects on peptide efficacy [17].

Several independent research groups have focused on peptidomimetic synthesis based on the α1-helix of hACE2. They have produced peptides ranging from 9–65 residues, and their work has highlighted the direct impact of helicity on antiviral activity [18,19,20,21,22,23]. However, a systematic comparative study of the *i*, *i* + 4 helix-inducing constraints leading to the desired hACE2 α-helical conformation has not been conducted to date.

Short (<9 AA) and long (>16 AA) Ala-based peptide sequences have exhibited helicities up to 71% and 80%, respectively, due to *i*, *i* + 3 and *i*, *i* + 4 side-to-side chain salt bridge interactions of the Glu with Lys/Arg [24,25]. These studies also pointed out that *i*, *i* + 4 stabilization is more effective than *i*, *i* + 3 stabilization; that finding is further supported by the formation of covalent lactam-based staple peptides between Asp/Glu and Lys that lead to 20-/21-membered rings [26,27]. Although their structure has been validated [26], the ubiquitous presence of peptidases limits the use of lactam bridges for in vitro and in vivo applications. Hydrophobic hydrocarbon staples obtained by ring-closing metathesis (RCM) have been optimized by systemically varying position, orientation, length and stereochemistry [28] to improve metabolic stability and cell permeability. As a result, enhanced bioavailability has been achieved [29,30]. In vivo efficacy was validated via the intravenous administration of *i*, *i* + 4 stabilized helical peptides bearing two S-pentenyl alanine residues (S5) in a murine model of cancer [31,32]. From a synthesis perspective, this technique offers on-resin cyclization and is orthogonal to amino acid-protecting groups but requires synthetically challenging α,α-disubstituted building blocks [33] to yield two double-bond isomers [34]. Because enhanced hydrophobicity can also lead to aggregation, a more hydrophilic thiol-ene stapling technique has recently been compared with hydrocarbon stapled analogues [35]. CD spectra measured in TFE/20 mM phosphate buffer for 16 residues peptides incorporating divinyl succinate staples yielded a comparative helicity of 23% relative to the hydrocarbon stapled control (25%) [36]. Another attractive alternative to increase the polarity of side chain crosslinking is via copper(I)-catalyzed Huisgen 1,3-dipolar azide-alkyne cycloaddition, which requires an inexpensive and less-toxic metal—copper—compared with ruthenium [37]. The “click” reaction provides great tolerance to functional groups and leads to a chemically stable 1,2,3-triazole that can participate in hydrogen bonding, dipole–dipole and π-stacking interaction, if positioned correctly [38,39]. The triazole staple strategy has been used to generate α- and 3_10_-helical peptides [37,38,40]. Double azide-alkyne cycloaddition between the synthetically prepared L-azidoalanine (L-Dap(N_3_)) residue and 1,5-hexadiyne (HDY) was successfully performed to create a 22-membered triazole staple peptide [39,41,42]. To achieve macrocycle size variability, a 23-membered thioether staple was synthesized via S-cysteine alkylation on a 1,4-Bis(bromomethyl)benzene (BMB) substrate [17]. Another method exploiting only inexpensive proteinogenic residues involved performing a cysteine perfluoroarylation with hexafluorobenzene (HFB), which resulted in an aromatic 1,4-regioselective, rigid and lipophilic linker. Such regio- and chemo-selective nucleophilic aromatic substitution (SNAr) can be carried out in solution [17,43] on unprotected peptides or on resin; the latter requires selective cysteine deprotection and produces high yields [44]. Together, the chemical tools currently available for cross-linker formation can confer a range of lengths, physicochemical properties and rigidities due to a variety of chemical transformations. They are accordingly worthy of a comparative investigation.

The incorporation of helix-inducing constraints, and thus the replication of protein-inspired α-helical interaction motifs, is of great interest in the design of pharmacological tools. However, analyzing the structure stabilization resulting from linkers has proven to be challenging, largely due to variabilities in peptide sequences and lengths and the limited availability of consistent experimental data pertaining to the helicity of short peptides. In this study, we systematically evaluated the impact of various *i*, *i* + 4 cyclizations on two short linear hACE2-derived sequences that adopt a random-coil structure in solution. The most helical peptides identified via CD were further characterized using NMR and two distinct neutralization assays in order to evaluate the consequences of the conformational constraints imposed. Our results provide guidance for selecting an optimal staple strategy in the de novo design of short protein-inspired 3.6_13_-helical peptides, based on structural recognition of “hot spot” residues.

## 2. Results and Discussion

### 2.1. Experimental Design and Synthesis

The first sequencing of the SARS-CoV-2 genome [45] led to a 3.5 Å (and 3.2 Å) resolution cryogenic electron microscopy (Cryo-EM) reconstruction of the Spike (S) trimer in the prefusion state with a single receptor binding domain (RBD) in up conformation [46,47]. The SARS-CoV-2 S RBD protein bound to ACE2 was resolved using cryo-EM and X-ray crystallography shortly thereafter [1,2,48]. Other groups focused on the structural resolution of the SARS-CoV-2 S RBD protein in the presence of full-length hACE2 or a neutralizing antibody: CR3022 [49,50]. These early and highly accurate [51] structural insights provided a blueprint for our rational structure-based design of peptidomimetics. 

To compare the intrinsic similarity of the SARS-CoV-2 RBD domain bound to hACE2, we explored four models with Protein Data Bank (PDB) access codes 6M0J, 6VW1, 6LZG and 6M17. The models were selected on the basis of their diversity of techniques, their resolution and the specific target proteins they investigated. Our detailed PPI analysis revealed 17 residues within hACE2 involved in hydrogen bonds, salt bridges, aromatic-aromatic and hydrophobic interactions with the SARS-CoV-2 S RBD (Appendix A). Within all four resolutions, only three residues—Asp38, Tyr83 and Lys353—shared the same interacting partner (Tyr449, Asn487 and Gly496, respectively). These subtle differences that we observed when we analyzed the four resolved structures suggest that the SARS-CoV-2 RBD/hACE2 interface is highly complementary and dynamic in nature. Furthermore, it is difficult to assign a significant importance to only one particular residue. Ten amino acids contributing to the H-bonding network are part of a highly organized 3.6_13_-helix (α-helix) of hACE2, characterized by a main chain N-H_(*i*)_∙∙∙O=C_(*i*+4)_ bonding pattern (Appendix A). We accordingly exploited the compact and continuous α1-helix (Ser19-Thr52) of hACE2 to ensure that our PPI inhibitors were able to achieve the best possible target recognition. We note that none of the previously described rare (<1%) hACE2 variants included a mutation (K26R, D206G G211R, R219C, K341R, I468V) in the studied contact area of SARS-CoV-2/hACE2 [52], which would have impacted our rational peptide design.

Using an in silico alanine scan (BudeAlaScan) performed on the α1-helix of hACE2 taken from the crystal structure of SARS-CoV-2 S RBD bound with ACE2 (PDB 6M0J), we identified residues critical to binding (Figure 2). Of the 34 amino acids, 16 are important energetic contributors to the overall interface energy. Thr27 and Leu45 face a hydrophobic binding pocket at the SARS-CoV-2 interface and were therefore considered less critical in our design (Appendix A). On the other hand, Asp30 and Tyr41 exhibited the highest energetic contributions (8.8 and 12.3 kJ/mol, respectively). We therefore deleted 11 residues from the C-terminal region and 10 residues from the N-terminal region of α1-helix (Ser19-Thr52). Doing so produced two short mid-section sequences: Asp30-Asp38 and His34-Gln42.

The 3.6_13_-helical secondary structures are naturally stabilized by hydrogen bonds between the amide nitrogen and the carbonyl oxygen atoms positioned *i*, *i* + 4 but destabilized by unfavorable C- and N-terminal charges through the “charge-helix dipole” interaction [53]. Therefore, we introduced the acetyl group at the N-terminus and the carboxamide group at the C-terminus of the staple peptides. We accordingly created additional internal hydrogen bonds at both ends. 

Moreover, each additional salt bridge formed by Glu/Asp and Lys/Arg [24] or Gln at position *i* + 4 [54] may substantially contribute to the helical content of short peptides. With respect to the residues involved in the PPI between SARS-CoV-2 Spike RBD/hACE2 and potential salt bridges, we aimed to stabilize the α-helical conformation of hACE2 analogues by substituting the non-essential amino acids at positions *i* and *i* + 4. Doing so resulted in two sequences, *Ac*-NH-Asp-Lys-X_1_-Asn-His-Glu-X_2_-Glu-Asp-CO-*NH_2_* and *Ac*-NH-His-Glu-X_1_-Glu-Asp-Leu-X_2_-Tyr-Gln-CO-*NH_2_*, where X_1_ and X_2_ correspond to amino acids present in the natural hACE2 protein or to variable linker constituent residues. We note that the second sequence identified (His34-Gln42) contains an additional “hot spot” residue compared with the hACE2 sequence (Asp30-Asp38).

We also obtained linear derivatives **1** and **2** of the native hACE2 sequence. We synthesized covalent lactam-based peptides **3**–**8**, **22** and **23** on resin using orthogonally protected lysine by allyl carbamate and aspartic/glutamic acid by allyl ester. We used structurally less flexible α,α-disubstituted amino acids (S5) to obtain hydrocarbon staple compounds **9**, **10**, and **11** with a characterized isomer configuration. Analogues **14**–**17** exhibited a triazole cross link using the bio-orthogonal “Click” reaction, which was recently the subject of a Nobel prize in chemistry [55,56,57]. To fully explore the effect of physicochemical bridge properties on helicity, we obtained thioether staple peptides **18** and **19** by S-alkylation to form 23-membered rings. Alternatively, peptides containing *i*, *i* + 4 disubstituted cysteines yielded 21-membered conformationally constrained peptides **20** and **21** via SNAr on an electron-withdrawing, group-activated ring such as HFB. In total, we synthesized various helix-inducing constraints contributing to the overall polar or hydrophobic surface of the staple peptides given our goal of better understanding conformation and biological activity (Figure 3B,D).

### 2.2. Circular Dichroism

The Far-UV CD spectra revealed that linear peptides **1** and **2** derived from the α1-helix of hACE2 (Asp30-Asp38) and (His34-Gln42), respectively, folded into a random-coil structure that exhibited characteristic small dichroism at 222 nm and stronger negative dichroism at roughly 200 nm (Figure 3A,C). We confirmed these results using the online analytical tool AGADIR [58], which predicted helicity in the conditions used to generate the CD experimental data (i.e., pH 7.4 and 25 °C) (Figure 3B,D). In the presence of 50% TFE [59], both linear peptides exhibited a higher helical content of up to 36.9% and 27.3%, respectively (Appendix A). This indicates their steric potential to adopt an α-helical conformation.

The Far-UV CD spectra recorded for the hACE2-derived staple peptides (Asp30-Asp38) did not display the characteristic shape expected for an α-helical conformation (i.e., one maximum between 190 and 195 nm and two negative minima at approximately 208 and 222 nm, as previously reported [60]). However, our analysis of the Far-UV CD spectra of the hydrocarbon staple peptide **10** with BeStSel (online software for CD spectral analysis) revealed the presence of an α-helical character to some extent (28.9%); we noted a significant population of an antiparallel, right-twisted β-sheet conformation (23.2%). Proline residue at the N-terminus has also been suggested to induce helicity [53]. This finding prompted us to synthesize the lactam stapled peptides **22** and **23**. Similar to our observations of compound **10**, we also noted a dominant conformation of **22** attributable to a regular helix1 (53.7%) and an antiparallel, right-twisted β-sheet (46.3%). Therefore, the high percentage of calculated helicity in compounds **10** and **22** can be explained on the basis of the contribution of a strong, negative absorbance band within the 217–218 nm spectral range of the Far-UV CD spectra. Such features are typical of a right-twisted β-sheet [60,61].

We were also successful at applying several cyclization strategies across the panel of hACE2 (His34-Gln42)-derived peptides. Lactam **4**, **23** and hydrocarbon **11** stapled compounds not only displayed an unprecedented degree of helicity (up to 55.5%, 56.7% and 53.5%, respectively), but peptides **4** and **11** also exhibited the highest attainable helicity for such short sequences (Appendix A). Moreover, the CD spectra obtained for linear and respective cyclic compounds revealed that a stabilized helical conformation is associated with the selected staple strategy (Appendix A). BeStSel analyses demonstrated that the helical content of **4** was 37.0%; 18.7% was ascribed to a regular helix1, and 18.3% was ascribed to a distorted helix2. The associated root mean square deviation (RMSD) was 0.5. As expected, **11** exhibited a higher helical content of 41.9% with BeStSel compared with **4** and **23**; 24.2% was attributed to a regular helix1. The distortion of the helical conformation can be partially explained by the highly polar nature of our peptide sequence (7 out of 9 amino acids) and inherent limitations in terms of the maximum helicity possible for short peptides due to C- and N-terminal distortions. Unlike hACE2 derivatives (Asp30-Asp38), the double triazole staple cyclization technique applied to the hACE2 sequence (His34-Gln42) resulted in **17** exhibiting a helicity of 50.7%. That finding highlights the substantial influence of the initial linear sequence on staple-induced helicity. 

Lactam and hydrocarbon staple peptides exhibited higher helicity than linear or other staple techniques. We next conducted analyses using NMR.

### 2.3. Nuclear Magnetic Resonance

#### 2.3.1. ^1^H Resonance Assignments

Whereas CD spectro-polarimetry can provide an estimate of the α-helical content of a peptide, NMR enables the determination of its precise location and extent. ^1^H-^1^H-Total Correlation Spectroscopy (TOCSY) (mixing time = 50 ms) and ^1^H-^1^H-Nuclear Overhauser Effect Spectroscopy (NOESY) (mixing time = 300 ms) were used to assign ^1^H resonances for lactam **4** and hydrocarbon **9**, **10**, **11** staple peptides (Appendix A, Table 1 and Appendix A). Except for the Gln at the C-terminus of hACE2-derived sequences (His34-Gln42), NH, CαH and CβH were unambiguously assigned using the classical sequential assignment procedure.

#### 2.3.2. NMR-Based Peptide Structural Comparisons and Hydrocarbon Staple Isomeric Investigations

Because compounds **4** and **11** share the same linear sequence and differ only in the nature of their staple, we rigorously compared their NH and CαH chemical shifts. Surprisingly, we found that only the NH and CαH protons of the Leu residue preceding the linker were significantly upfielded in compound **11** compared with compound **4**. We can only assume that this finding arises from steric hindrance of Me-Cα that in turn results in local distortion from helical structure and/or simply affects the electron density around Cα. We note that the methyl group has been previously reported to promote proteolytic stability [17].

The RCM reaction performed on *i*, *i* + 4 S5-substituted hACE2 derivative (Asp30-Asp38) yielded two products **9** and **10** as a mixture of E- and Z- isomers with an E:Z ratio of 5:3 in moderate yields. (The E:Z ratio determined by isolated yields.) The E:Z selectivity was characterized using the coupling constants ^3^J_CH=CH_ = 15.6 Hz and 9.6 Hz between two olefinic protons of peptides **9** and **10**, respectively (Appendix A). Peptide **9** exhibited a conformation that differed from that of the linear native peptide **1** and was therefore of interest for further characterization with TOCSY. 

Better selectivity was achieved for the *i*, *i* + 4 S5-substituted, hACE2-derived peptide (His34-Gln42) (E:Z = 13:3), which enabled peptide **11** to be isolated with a high purity and furthermore characterized using the coupling constant ^3^J_CH=CH_ = 16.2 Hz between two olefinic protons (Appendix A). Compound **11** attained a helical content of up to 53.5% based on CD. This finding suggests clear superiority of the TRANS isomer in terms of helicity but only due to the favorable entropic conformational flexibility of the double bonds in the RCM reaction context. These results also indicate that the initial sequence has a strong impact on the selectivity of the RCM reaction, which is consistent with previously published results [34].

#### 2.3.3. Secondary Chemical Shifts of Peptides **4**, **9**, **10** and **11**

We next exploited the chemical shifts of the CαH and NH protons as indicators of secondary structure. We compared the ^1^H NMR assignments with the characteristic random-coil ^1^H chemical shifts previously reported in the literature [62]. As expected, all of the CαH protons in peptide 4 were found to be upfield shifted from the random-coil value by −0.22 ppm on average. That finding is indicative of helical conformation throughout the entire peptide (Figure 4). The largest upfield shifts were observed for the amino acids Lys_(linker)_, Glu_2_, Asp and Leu, which are encompassed by the staple or form the linker themselves. That result is suggestive of a strong helical constraint induced by the lactam bridge.

The secondary (−∆δ) NH and CαH proton chemical shifts of hydrocarbon staple peptides **9**, **10** and **11** are much smaller than those of peptide **4** (Appendix A). This finding reveals that peptides **9**, **10** and **11** are less likely to adopt a helical conformation compared with **4**. Peptide **11** experiences a strong end effect given that the His and Tyr residues at the N-terminus and C-terminus, respectively, are strongly downfielded by −0.23 ppm on average from the random-coil value (Appendix A).

This end effect, which is not observed for the lactam staple peptide, can be explained by the long side chain of the Lys_(linker)_ residue. That chain forces the linker to adopt a highly constrained conformation by bringing the NH and CαH protons of the terminal residues into an ideal position [26]. The data presented above show that a lactam bridge is the best strategy to stabilize the helical conformation of short peptides derived from hACE2 sequence (Asp30-38).

#### 2.3.4. α-Helical NOESY Connectivities of Peptide **4**

Ideally, ^1^H two-dimensional NOESY spectra of an α-helical peptide enable the assignment of sequential short-range d_NN(i,i+1)_ and d_αN(i,i+1)_, as well as sequential and medium-range d_αN(i,i+3)_ and d_αN(i,i+4)_. However, only short-range nuclear Overhauser effect (NOE) could be observed. This finding is due to NOE’s low efficacy for such short peptides; the correlation times fall in between the extreme narrowing and spin-diffusion limits (Figure 5).

Resonance assignments within the amide d_NN_ and the fingerprint d_αN_ regions of the NOESY spectrum of the lactam stapled peptide **4** that enable secondary structure characterization are shown in Figure 4. Strong sequential d_NN(i,i+1)_ connectivities are observed between the amino acids Lys_(linker)_, Glu_2_, Asp, Leu, Asp_(linker)_ and Tyr of peptide **4** (Figure 5A,B,D). This finding reveals the helix-stabilizing effect of the lactam staple. Accordingly, the α-proton chemical shifts of these residues are upfield shifted relative to the random-coil value (Figure 4). Consistent with the chemical shift index, this finding is an additional validation of the α-helical conformation of these residues. Moreover, as expected, we observed stronger d_NN(i,i+1)_ and weaker d_αN(i,i+1)_ connectivities on average (Figure 5D). 

The apparent coupling constants (^3^J_NH-αCH_ in Hz) for peptide **4** all lie in the range of 3 Hz ≤ ^3^J_αNH-αCH_ ≤ 7 Hz, with an average value of 5.5 Hz (omitting the unassigned Gln residue) (Figure 5D). Given that the ^3^J_αNH-αCH_ coupling constant is inferior to 6 Hz, we can only assume that peptide **4** exhibits a helical conformation with a dihedral angle φ of −57° ≤ φ ≤ 60°, which is characteristic of α-helix or 3_10_-helix [63]. We note that it is impossible to make a distinction between the two helical conformations given the relatively consistent values of ^3^J_αNH-αCH_ = 3.9 Hz for α-helix and ^3^J_αNH-αCH_ = 4.2 Hz for 3_10_-helix [63].

The d_αN(i,i+2)_ cross-peaks must be analyzed or further quantitatively evaluated to accurately distinguish between the α- and 3_10_-helical structure formed by the peptide. Generally, only the 3_10_-helix displays d_αN(i,i+2)_ connectivities [64]; the characteristic d_αN(i,i+2)_ distance for the α-helix is 4.4 Å and 3.8 Å for the 3_10_-helix [65]. Whether due to the very weak signal intensity or the need for more accurate structure determination, it must be stipulated that the measured NOESY spectra of peptide **4** lack d_αN(i,i+2)_ cross-peaks and that therefore neither peptide **4** (nor **9**, **10** and **11**) display any 3_10_-helical characteristics along their extents. 

#### 2.3.5. Comparative NOESY Connectivities for RCM-Based Staple Peptides

To investigate the influence of linear sequences and isomer formation on conformation in RCM-based staple peptides, we compared the NOESY spectra of peptides **9**, **10** and **11** (Appendix A). The reduced number of d_NN_ connectivities in peptides **9** and **10** indicates a lower conformational rigidity compared to that of peptide **11**. Peptide **10** acquired additional d_αN(i,i+1)_ and d_αN(i,i+2)_ connectivities compared to peptide **9**, but that increase alone was not sufficient to favor the CIS isomer.

As in peptide **4**, we measured stronger d_NN(i,i+1)_ and weaker d_αN(i,i+1)_ connectivities for peptide **11**; this finding indicates the adoption of a helical conformation (Appendix A). The NOESY spectrum of peptide **11** (Appendix A) exhibits significantly greater solvent exchange of the peptide’s backbone amide protons compared with peptide **4**. That result suggests that peptide **11** is prone to movement. 

Our observations of upfielded α-proton chemical shifts, coupling constants and NOESY connectivities between protons of neighboring and non-neighboring amino acid residues are all consistent: they provide independent, indirect and direct evidence that peptide **4** adopts an α-helical conformation in solution throughout its entire chain, with the exception of the more-flexible terminal residues.

### 2.4. Inhibition Assays

We evaluated the PPI disruption capacity of linear native peptides and staple analogues using a bioluminescence-based bioreporter assay. SARS-CoV-2 Large BiT-RBD (LgBiT-RBD) was co-incubated with the compounds for 30 min; we next added the Small BiT-ACE2 (SmBiT-ACE2) reporter for 5 min. The substrate, coelentrazine, was next added to determine the disruption capacity of analogues in relative luminescence units (RLUs) (Figure 6A).

As previously reported [66], an anti-SARS-CoV-2 Spike Glycoprotein RBD monoclonal antibody (NR-53795) was used as a positive control for biosensor validation and induced a complete loss of luminescence at 7.41 µM. Pre-incubation of LgBiT-RBD with linear peptides **1** and **2** of the native hACE2 sequence reduced the luminescence signal; a similar outcome was noted with lactam **4** and hydrocarbon **11** staple peptides. These results revealed that our compounds are capable of competing with hACE2 and disrupting SARS-CoV-2 Spike RBD/hACE2 interactions. 

However, we did not observe any statistically significant differences between the linear and staple peptides. That finding suggests that conformational changes in the linear analogues are responsible for the correct orientation of the side chain residues required for short-term interaction with RBD, which in turns leads to a reduction in luminescence.

To evaluate the long-term binding capacity and antiviral efficacy of hACE2 derivatives, we co-incubated staple peptides and their respective linear derivatives with SARS-CoV-2 pseudovirus for 30 min at 37 °C. We next conducted an additional 48 h of incubation with hACE2-HEK293T cells overexpressing transmembrane serine protease 2 (TMPRSS2). Evidence of efficient SARS-CoV-2 virus entry was confirmed by an increase in fluorescent expression upon cell infection. Conversely, inhibition of virus entry was observed when fluorescent expression was reduced compared to pseudovirus-only conditions, which was normalized to 100% infection. Our results are represented as the efficacy of viral entry inhibition (Figure 6B). The anti-SARS-CoV-2 Spike Glycoprotein RBD monoclonal antibody (NR-53795), which has previously demonstrated high neutralization potency with an IC_50_ of 0.11 µg/mL, was used as a positive control.

The lactam **4** and hydrocarbon **11** staple peptides inhibited 48% and 68% of the pseudovirus, respectively. Both compounds displayed enhanced inhibition compared with their respective linear sequences; compound **11** demonstrated a six-fold higher efficacy than that of **linear 11**. The reduced pseudoviral inhibition exhibited by the respective linear sequences shows that retention of conformation by peptide stapling at positions 36 and 40 is crucial for the long-term disruption of SARS-CoV-2 RBD/hACE2 interactions and, consequently, for neutralizing activity. Surprisingly, the hydrocarbon staple peptide **11** demonstrated a significantly higher inhibition efficacy than peptide **4**, which suggests that an excessive level of rigidity may be detrimental to its antiviral activity. In fact, a recent study demonstrated significantly superior antiviral activity of single-stapled peptides compared with their less-flexible, double-stapled analogues [21].

### 2.5. Plasma Stability

We determined resistance to proteolysis for linear **1** and **2**, lactam **3** and **4** and hydrocarbon **10** and **11** staple peptides by incubating the compounds for 24 h in rat plasma at 37 °C and recording degradation using UPLC-MS (Figure 7). As a reference control, we used a well-characterized 13-mer peptide known as Apelin-13 [67], which possesses a short in vitro half-life (t_1/2_). Additionally, Angiotensin II (AngII), an upregulator of hACE2 expression in human bronchial cells [68], is an important factor in coronavirus disease (COVID-19) lung infections and has a half-life of 4.2 h in rat plasma. Linear peptides **1** and **2** both exhibited levels of degradation similar to those of 8-amino acid-long AngII.

Interestingly, the lactam **4** and hydrocarbon **11** staple peptides were significantly more stable than their respective natural linear sequences after 24 h. That finding suggests that a helical conformation shields amide bonds within the peptide core and thereby prevents access to proteases. As expected, the in vitro half-lives of lactam staple peptides **3** and **4** (2.9 and 5.0 h, respectively) were slightly lower than those of RCM-based compounds **10** and **11** (4.0 and 6.8 h, respectively).

## 3. Materials and Methods

### 3.1. Reagents

Fmoc-protected amino acids, trifluoroacetic acid (TFA) and N,N-Diisopropylethylamine (DIPEA) were purchased from Combi Blocks (San Diego, CA, USA), Chem-Impex (Wood Dale, IL, USA) or Matrix Innovation (Quebec, QC, Canada). (S)-N-Fmoc-α-(4-pentenyl)alanine and Grubbs 1st Catalyst were obtained from Sigma Aldrich, St. Louis, MO, USA. Palladium (O)tetrakis(triphenyphosphine) (Pd(PPh_3_)_4_) was purchased from Strem Chemicals, Newburyport, MA, USA. TentaGel™ S RAM resin was purchased from Rapp Polymere, Tübingen, Germany. N,N-Dimethylformamide (DMF) and diethyl ether (Et_2_O) were purchased from Fischer Scientific, Hampton, NH, USA. All other reagents were purchased from Sigma Aldrich.

SARS-CoV-2 (2019-nCoV) Rabbit Mab neutralizing antibody was obtained from Sino Biological, Beijing, China (Catalog No. 40592-R001, HA140C2601). All media and supplements for cell maintenance were obtained from Life Technologies Ltd. (Paisley, UK), except otherwise stated.

### 3.2. Peptide Synthesis

#### 3.2.1. Solid-Phase Peptide Synthesis

All of the linear peptides were synthesized on a 0.1 mmol scale using a Symphony X^®^ Peptide Synthesizer from GYROS PROTEIN Technologies (Uppsala, Sweden) using TentaGel™ S RAM resin with a loading capacity of 0.23 mmol/g. Coupling of proteinogenic Fmoc-protected amino acids (5 eq, 0.2 M in DMF) was achieved by treatment with DIC (5 eq, 0.5 M in DMF) and Oxyma Pure (5 eq, 0.5 M in DMF) at room temperature for 45 min. Non-proteinogenic Fmoc-protected acids such as (S)-N-Fmoc-α-(4-pentenyl)alanine were double coupled for 1 h. Fmoc-protecting group was removed using a solution of 20% piperidine in DMF for 10 min. This step was repeated twice. After each coupling and deprotection reaction, the resin was washed with DMF five times. After the linear sequence was completed, the peptides were washed with DCM five times according to the standard procedure recommended by the Symphony X^®^ Peptide Synthesizer.

#### 3.2.2. N-Terminal Acetylation

After Fmoc deprotection with a solution of 20% piperidine in DMF, all of the peptides were treated on resin with acetic anhydride (3 eq), DIPEA (4.5 eq) and DMF (7 mL for 0.1 mmol of resin) for 20 min on an orbital shaker operating at 120 rpm. The resin was then washed with DMF (3 × 5 mL), DCM (3 × 5 mL), iPrOH (1 × 2 mL) and DMF (3 × 5 mL) prior to other reactions or peptide cleavage and global deprotection.

#### 3.2.3. Lactam Staple Formation

Peptides containing Fmoc-Lys(Alloc)-OH and Fmoc-Asp(OAll)-OH/Fmoc-Glu(OAll)-OH were treated on-resin to selectively remove the allyl carbamate of lysine and the allyl ester of aspartic/glutamic acid prior to on-resin side-to-side chain lactamization. Pd(PPh_3_)_4_ (0.1 eq) and N,N-dimethylbarbituric acid (4 eq) were pre-mixed in anhydrous DCM (2 mL for 0.1 mmol of resin) and added to the resin. The resultant solution was stirred at 25 °C for 1 h. The reaction was monitored with UPLC-MS after cleavage of the peptide from a resin aliquot. This procedure was repeated twice if we detected acetylated orthogonally protected linear peptide. The resin was then washed with DCM (2 × 5 mL), DMF (2 × 5 mL), 0.5% diethyldithiocarbamate in DMF (2 × 5 mL) and DMF (2 × 5 mL). Lactam staple formation was performed on-resin using BOP (1.5 eq) and DIPEA (2 eq) in 2:8 DMSO/NMP (2 mL for 0.1 mmol of resin). The reaction progression was monitored by using the Kaiser test [69]. BOP (1.5 eq) and DIPEA (5 eq) in a 2:8 DMSO/NMP ratio (2 mL for 0.1 mmol of resin) were added to the resin when we observed a positive (i.e., blue) Kaiser test; the resin was then agitated overnight at room temperature. If the Kaiser test was negative (i.e., yellow), the resin was washed with DMF (3 × 5 mL), DCM (3 × 5 mL), iPrOH (1 × 1 mL), DCM (3 × 5 mL) and DMF (3 × 5 mL); peptide cleavage/global deprotection, purification and characterization were then conducted according to the procedures described below.

#### 3.2.4. Hydrocarbon Staple Formation

Peptides containing (S)-N-Fmoc-α-(4-pentenyl)alanine were stapled on-resin via RCM. Following solid-phase peptide synthesis (SPPS) and subsequent acetylation, the resin was suspended in dry DCE (10 mL for 0.1 mmol of resin), filtrated and weighted out in microwave flash prior to the addition of Grubbs 1st Catalyst (0.2 eq) at 25 °C. The solution was stirred at 70 °C for 1 h in the microwave. We monitored the reaction via UPLC-MS after cleavage of the peptide from a resin aliquot. If we detected acetylated linear peptide, the resin was washed with DCE (3 mL), and we then performed a second round of RCM with fresh catalyst. The resin was then washed with DMF (2 × 5 mL), DCM (2 × 5 mL), iPrOH (1 × 5 mL), DCM (2 × 5 mL) and DMF (2 × 5 mL) prior to peptide cleavage/global deprotection, purification and characterization.

#### 3.2.5. Triazole Staple Formation

A copper(I)-catalyzed Huisgen 1,3-dipolar azide-alkyne cycloaddition for triazole staple formation [38] was performed on pure lyophilized peptide with the goal of accurately following up retention-time changes using UPLC-MS. A solution of linear peptide (1 eq) and CuSO_4_·5H_2_O (4.4 eq) in 2:1 H_2_O/t-BuOH (1 mL/mg peptide) was degassed with nitrogen for 15 min, and we then slowly added sodium L-ascorbate (4.4 eq) dissolved in H_2_O (2 mL). The reaction mixture was vigorously stirred in a nitrogen atmosphere for 30–90 min at room temperature. After removing the t-BuOH in vacuo, the crude was lyophilized and then purified by preparative HPLC-MS.

#### 3.2.6. Double Triazole Staple Formation

A copper(I)-catalyzed Huisgen 1,3-dipolar azide-alkyne cycloaddition for double triazole staple formation was performed on crude linear di-azido peptide sequences; we adopted a procedure similar to that reported in the literature [39]. A solution of diazido peptide (1 eq) and dialkynyl linker (1.1 eq) in a 1:1 H_2_O/t-BuOH ratio (1 mL/mg peptide) was degassed with nitrogen for 15 min, followed by the addition of CuSO_4_·5H_2_O (1 eq), tris(3-hydroxypropyltriazolylmethyl)amine (1 eq) and sodium L-ascorbate (3 eq). After stirring the solution in a nitrogen atmosphere at room temperature for 24 h, the reaction mixture was lyophilized and purified via preparative HPLC-MS to yield the final stapled peptide. If the analytical UPLC exhibited a broad peak near the expected stapled peptide peak, we added EDTA (1 eq) to the lyophilized crude prior to purification.

#### 3.2.7. Thioether Staple Formation (S-Alkylation)

Thioether staple was performed on purified linear di-Cysteine sequences according to a previously described protocol [17]. We added 1,4-Bis(bromomethyl)benzene (1.5 eq) to a solution of purified lyophilized peptide (1 eq) in 100 mM Tris solution (1 mL/mg peptide) at 25 °C; the resulting solution was stirred until complete conversion was attained and confirmed via UPLC-MS. Then, the reaction mixture was acidified to pH 4 via the addition of a solution of 0.1% TFA. The crude was then lyophilized and purified by preparative HPLC-MS to yield the final stapled peptide.

#### 3.2.8. Thioether Staple Formation (S-Arylation)

Di-substituted cysteine-containing peptides were stapled by SNAr on HFB using a procedure similar to that reported previously in the literature [44]. After SPPS and subsequent acetylation, the resin was washed with DCM (2 × 5 mL) and DMF (2 × 5 mL). We next applied on-resin selective deprotection of trityl-protected cysteine residues using a solution of TIPS/TFA/DCM (1/4/35, 2 mL/10 mg of resin), for 1 h at 25 °C on an orbital shaker at 120 rpm. The resin was subsequently filtered to discard the cleavage solution and washed with DCM (2 × 5 mL) and DMF (2 × 5 mL). Next, a solution of HFB (10 eq) and DIPEA (0.4 M in DMF; 1 mL/10 mg of resin) was added to the swollen peptide resin and shaken at 25 °C overnight on an orbital shaker at 120 rpm. Subsequently, the reaction mixture was gently evaporated under a stream of weak airflow and the resin was washed with DMF (2 × 5 mL), DCM (2 × 5 mL), iPrOH (1 × 5 mL), DCM (2 × 5 mL) and DMF (2 × 5 mL) prior to peptide cleavage/side chain deprotection, purification and characterization.

#### 3.2.9. Peptide Cleavage and Side Chain Deprotection

Peptides were cleaved from the resin using a solution of TFA/TIPS/H_2_O (95/2.5/2.5, 1 mL/30 mg of resin) for 2 h on an orbital shaker at 120 rpm. The resin was subsequently filtered, and the peptide was precipitated in cold Et_2_O. The suspension was centrifugated at 4 °C for 10 min at 2000 rpm in an Allegra TM 6R Centrifuge (Beckman Coulter Life Sciences, Indianapolis, IN, USA). Finally, the supernatant was decanted into the waste, and the remaining white solid was dissolved in a mixture of H_2_O and ACN with 0.1 % TFA and lyophilized overnight in a freeze dryer (LABCONCO, Kansas City, MO, USA).

#### 3.2.10. Peptide Purification

The previously lyophilized peptide was dissolved in a 1.8 mL mixture of H_2_O and ACN depending on peptide solubility and filtrated through a 0.22 µm PTFE Chromatography Syringe Filter. The peptides were purified using Waters preparative HPLC-MS (column XSELECTTM CSHTM Prep C18 (19 × 100 mm) packed with 5 μm particles, UV detector 2998, MS SQ Detector 2, Sample manager 2767 and a binary gradient module). We isolated peptides with a purity exceeding 95% for use in subsequent experiments (Appendix A).

### 3.3. General Procedures

#### 3.3.1. Peptide Characterization and Data Analysis

All of the data were analyzed using GraphPad Prism (version 9.3.1) (San Diego, CA, USA).

##### UPLC-MS Analysis

All of the peptides were analyzed using a Waters UPLC-MS system (column Acquity UPLC^®^ CSHTM C_18_ (2.1 × 50.0 mm) (Agilent Technologies, Santa Clara, CA, USA) packed with 1.7 µm particles). A 5–95% gradient of ACN and H_2_O containing 0.1% TFA was used over the course of 2.5 min at a flow rate of at 1 mL/min at 25 °C.

##### HRMS Analysis

High-resolution mass spectra were acquired on a maXis 3G orthogonal mass spectrometer (ESI-QqTOFMS) (Bruker Daltonik; Bremen, Germany) using electrospray ionization in positive (or negative) ion mode. Each compound was dissolved in dichloromethane (or methanol) and then diluted 10-fold in methanol. Each solution was then infused individually into the ESI-QqTOFMS using a syringe pump at a flow rate of 5 µL/min. Mass spectra were recorded over a range of *m*/*z* 50–1200, and external calibration was performed using a sodium formate 0.5 mM solution.

#### 3.3.2. Peptide Plasma Stability

At each time point—i.e., 1, 2, 7 and 24 h—0.2 mM peptide (6 µL) was incubated in 27 µL of rat serum at 37 °C with agitation. Then, the 96-well PTFE Chromatography Syringe Filter plate was cooled to room temperature, and the serum was precipitated using a solution containing 0.1% formic acid and 0.25 mM N,N-dimethylbenzamide (internal standard) in 50/50 ACN/EtOH (140 µL). We used a multichannel pipette to collect 173 µL of each precipitate, which was then transferred to an Impact™ protein-precipitation plate (Phenomenex Inc., Torrance, CA, USA) assembled with a 96-well plate and centrifuged at 25,000 rpm at 4 °C for 10 min. We then retrieved 5 µL of the filtrate from the 96-well plate and subjected it to UPLC-MS analysis. A 5–95% gradient of ACN and H_2_O containing 0.1% TFA was used over 2.5 min at a flow rate of at 1 mL/min at 25 °C. Each integration of the peptide peak was normalized over the internal standard integration at the same time point, *t* = *x*. Proteolytic degradation was quantified as a ratio of the normalized value at *t* = *x* relative to the normalized value for the peptide at *t* = 0, using Equation (1):(1)$$\frac{\frac{{Compound\;integration\;}_{t=x}}{{Internal\;standard\;integration\;}_{t=x}}}{\frac{{Compound\;integration\;}_{t=0}}{{Internal\;standard\;integration\;}_{t=0}}}\times 100=\%\;of\;compound\;degradation.$$

Each experiment was repeated in duplicate for each time point for each peptide.

#### 3.3.3. Circular Dichroism Spectroscopy

##### Circular Dichroism Measurements

CD measurements were performed in a quartz flow cell with a 1 mm path length at room temperature (25 °C) using a Jasco J-810 CD spectrometer. We collected spectra over a wavelength range of 190–250 nm at a speed of 50 nm/min. The bandwidth was 1 nm, we averaged over 10 scans and the baseline (phosphate-buffered saline (PBS) only) was subtracted from each spectrum. The peptides were dissolved in PBS (pH 7.4) at a concentration of approximately 100 µM (0.1–1.0 mg/mL).

##### Quantification of Helicity

The raw CD data (mdeg) were converted to mean residue ellipticity ([θ], deg·cm^2^·dmol^−1^·res^−1^) by normalizing for peptide concentration, the number of amide bonds and path length, using Equation (2):(2)$$[\theta ]=\frac{\theta }{10\times C\times N_{p}\times l}$$
where *θ* is the measured ellipticity (mdeg), *C* is the peptide molar concentration (M), *N_p_* is the number of amino acid residues in the peptide and *l* is the cell path length (cm).

The maximum mean ellipticity [*θ*]*max* was calculated using Equation (3), as reported previously:(3)$${\left[\theta \right]}_{max}=\left(-44000+\left(250\times T\right)\right)\times \left(1-\frac{k}{N_{p}}\right)$$
where *T* is temperature (°C) and *k* is the number of non-hydrogen-bonded peptide carbonyls. For N-terminal acetylated peptides, *k* = 3. For N-terminal-free peptides, *k* = 4 because the ellipticity at 222 nm is mainly affected by the electronic environment of the peptide carbonyl [70].

Because the random coil and the maximum mean ellipticity are temperature dependent, Equation (4) must be used when calculating the percent helicity:(4)$${\left[\theta \right]}_{c}=2220-53\times T.$$

The percent helicity can then be calculated using Equation (5):(5)$$\%\;Helicity=\left(\frac{{\left[\theta \right]}_{222}-{\left[\theta \right]}_{c}\;}{{\left[\theta \right]}_{max}-{\left[\theta \right]}_{c}}\;\right)\times 100.$$

#### 3.3.4. Nuclear Magnetic Resonance Measurements and Data Analysis

The NMR samples were prepared in 10% D_2_O (Sigma-Aldrich, Poole, UK), 40 mM K_2_HPO_4_ pH 6.8, 40 mM KCl and 0.2% NaN_3_. ^1^H-^1^H NOESY spectra (mixing time of 300 ms), ^1^H-^1^H TOCSY (mixing time 50 ms) and ^1^D-^1^H spectra were acquired at 25 °C on a Bruker Avance operating at a ^1^H frequency of 600 MHz and equipped with a cryogenic probe (TCI H&F-C/N-D-05 2 XT) and Z-axis pulsed-field gradients. The NMR data were processed using Topspin 3.6.2 (Bruker) and analyzed using CcpNmr software (version 3.1.1) (Leicester, UK) or MestReNova chemistry software (version 12.0.0-20080) (©2017 Mestrelab Research S.L., Santiago de Compostela, Spain). Chemical shifts (d) are reported in parts per million (ppm).

#### 3.3.5. Bioreporter-Based Neutralization Assay

##### Cell Maintenance

Human embryonic kidney (HEK293) cells were cultured in DMEM supplemented with 10% Fetal Bovine Serum (FBS) and 1% penicillin/streptomycin at 37 °C in a 5% CO_2_ atmosphere.

##### Generation of Biosensors

HEK293T cells at 70% confluency were transfected with SmBiT-ACE2 or LgBiT-RBD using PolyJet transfection reagent (Signagen, MD, USA) following the manufacturer’s protocol. Forty-eight hours after transfection, the cells were lysed using 1× passive lysis buffer (Promega, Madison, WI, USA, Cat.#E1910) on ice and centrifuged to clear the lysate. Lysates that were not used immediately we restored at −80 °C.

##### Biosensor Assay

In an opaque, white 96-well plate, 50 μL of cell lysate containing biosensor LgBiT-RBD was co-incubated with 50 μL of compound that had been previously prepared at 1 mM in PBS (pH 7.4). Thirty minutes later, the cell lysate containing the biosensor SmBiT-ACE2 was added for 5 min. Meanwhile, the lyophilized native coelenterazine (Nanolight Technology, Pinetop, AZ, USA, Cat #303–500 μg) was resuspended in 100% EtOH to generate a 2 mM stock solution. A working reagent for the lysed cell assays was created by adding 5 μL of 2 mM stock solution to 1 mL of PBS (pH 7.4). Then, in an opaque, white 96-well plate, 50 μL of working coelenterazine substrate was added to each well immediately before reading the luminescence. A Mithras^2^ LB 943 Multimode Reader (MBI Lab equipment Montreal Biotech Inc., Quebec, QC, Canada) was used to measure the luminescence.

##### Bioreporter-Based Inhibition Assay Data Analysis

The inhibitory data were plotted in RLUs using GraphPad Prism (version 9.3.1). Each result constitutes three independent experiments, measured in triplicate each time, and the data are expressed as means ± SEM (error bar). The means of more than two groups were compared using one-way ANOVA with Tukey’s multiple-comparison correction. For all analyses, **** *p* < 0.0001;n.s., not significant.

#### 3.3.6. Neutralization Assay

##### Lentivirus Vector Design

pHAGE-CMV-Luc2-IRES-zsGreen was obtained from BEI Resources (Catalog No. NR-52516). The packaging vector psPAX2 was a gift from Didier Trono (Addgene plasmid #12260; http://n2t.net/addgene:12260 (19 April 2020); RRID:Addgene_12260). The SARS-CoV-2 expression vector (pcDNA3.1-Spike) was a gift from Gary Kobinger and expressed a codon-optimized version of the SARS-CoV-2 Spike protein (GenBank accession number MN985325.1). The Δ19 Spike was generated by PCR mutagenesis and consisted of a truncation of the last 19 amino acids of the SARS-CoV Spike protein for pseudotyping enhancement [71].

##### Cell Maintenance

Human embryonic kidney cells stably expressing the human Angiotensin Converting Enzyme 2 (HEK-293T-hACE2, BEI Resources Catalog No. NR-52511) and HEK 293T/17 (ATCC, CRL 11268) cells were maintained in DMEM supplemented with 10% FBS and 1% penicillin/streptomycin. All of the cells were cultured at 37 °C under a humid atmosphere containing 5% CO_2_.

##### Generation of Pseudoviral Particles

The 293T cells were cultured in DMEM supplemented with 10% FBS and 1% penicillin/streptomycin at 37 °C in a 5% CO_2_ atmosphere. Adherent HEK293T/17 cells (2,000,000 cells per Petri dish) were seeded in DMEM growth media 24 h prior to co-transfection. Twenty-four hours after seeding, the cells were co-transfected with plasmids required for pseudoviral production: 4 μg of pHAGE-CMV-Luc2-IRES-zsGreen (transfert vector), 3 μg of psPAX2 (packaging vector) and 3 μg of pcDNA3.1-SARS-CoV-2 spike (spike glycoprotein) using polyethyleneimine (PEI, 25K Polyscience 23966-1, Warrington, PA, USA). The cells were incubated at 37 °C in a 5% CO_2_ atmosphere for 48 h. Next, the pseudoviral particles were collected by harvesting the supernatant from each Petri dish; the particles were stored frozen at −80 °C.

##### Pseudovirus-Based Inhibition Assay

All of the compounds were dissolved in a stock solution of DMEM at 10 mM prior to the experiments and stored at −20 °C if they were not used immediately. HEK-293T-hACE2 cells were chemically transfected with 3 μg of pCMV3 hTMPRSS2 (Sino Biological HG13070-UT) plasmid. Following post-transfection, the cells were seeded at 36,500 cells/well in dark-bottomed, sterile 96-well plates the day prior to infection. 1.5 h prior to infecting the cells, all of the compounds were diluted from stock solutions in pre-warmed complete DMEM media. Then, compounds were pretreated with pseudoviral particles for 30 min and then incubated with HEK-293T-hACE2 cells transfected with TMPRSS2 for two days. Following incubation, the media was gently removed, and the cells were washed with HBSS three times and incubated for 30 min at 37 °C. Fluorescence measurements were performed using a TECAN plate reader (© 2023, Tecan Trading AG, Männedorf, Switzerland) at room temperature (excitation: 360 nm, emission: 460 nm).

##### Pseudovirus-Based Inhibition Assay Analysis

The fluorescence data (RFUs) were converted to percentages by normalizing them with the control without infection (DMEM) set as 0%; we presumed that the pseudovirus-only samples represented maximum infection (100%). We made use of GraphPad Prism (version 9.3.1). Each data point represents three independent experiments, measured in triplicate, and the data are expressed as means ± standard error of the mean (SEM) (error bar). The means of more than two groups were compared using one-way ANOVA with Tukey’s multiple-comparison correction. For all analyses, **** *p* < 0.0001; ** 0.0017 ≤ *p* ≤ 0.0025; n.s., not significant.

#### 3.3.7. Peptide Design and Visualization

The rational design of the peptides was based on previously published high-resolution structures: PDB 6M0J [1] (Crystal structure of SARS-CoV-2 S RBD bound with ACE2, X-ray diffraction, 2.45 Å, 18 March 2020), PDB 6VW1 [48] (Structure of SARS-CoV-2 chimeric RBD complexed with its receptor hACE2, X-ray, 2.68 Å, 4 March 2020), PDB 6LZG [2] (Structure of novel coronavirus S RBD complexed with its receptor ACE2, X-ray diffraction, 2.50 Å, 18 March 2020) and PDB 6M17 [49] (The 2019-nCoV RBD/ACE2-B0AT1 complex, electron microscopy, 2.90 Å, 11 March 2020).

Two helical peptide segments of hACE2 (residues Asp30 to Asp38 and His34 to Gln42) were taken from the crystal structure of SARS-CoV-2 S RBD bound with ACE2 (PDB code 6M0J) and used as a template for the staple peptides. Backbone amino acids at the position Phe31 and Ala36 (or Ala36 and Phe40) were substituted with different amino acids of varying lengths, orientations and stereochemistries before being stapled together with or without an additional linker by corresponding chemistries. Finally, energy minimization was performed using Molecular Operating Environment (MOE) (Chemical Computing Group, Quebec, QC, Canada).

## 4. Conclusions

This study explores a variety of helix-inducing *i*, *i* + 4 constraints exploiting lactam, hydrocarbon, triazole, double triazole and thioether bridges on hACE2-derived sequences, as well as the resulting conformations and PPI inhibitions.

In general, for both the hACE2-derived peptide sequences Asp30-Asp38 and His34-Gln42, the incorporation of an *i*, *i* + 4 lactam bridge formed of Lys (N-terminus) and Asp (C-terminus) stabilized a high percentage of helical conformation (up to 55.5%). The staple requirement proves undeniable: the respective linear sequences exhibited significantly lower helical contents. Moreover, the magnitude of helicity generated in 50% TFE for lactam- and hydrocarbon-based peptides was not any higher, which indicates that short sequences just nine amino acids in length attained maximum helical values. Our results also showed that the initial linear sequence of a peptide has a strong influence on the maximum helicity achievable. That finding likely arises due to the distribution of charged and/or hydrophobic residues.

Our NMR results were consistent with the CD spectra obtained for peptides **4** and **11**. That result supports our assertion that an α-helical conformation was adopted in solution. A comparison of individual CαH proton chemical shifts relative to the random-coil standard values revealed a considerable upfield shift across the entire peptide sequence. Additionally, the d_NN(i,i+1),_ d_NN(i,i+2)_ and d_αN(i+1)_ NOESY connectivities strongly supported the idea that residues forming, are constrained by or following the lactam bridge are in an α-helical conformation. Moreover, the presence of more intense d_NN(i,i+1)_ cross-peaks relative to d_αN(i+1)_ cross-peaks provided further indirect evidence that peptide **4** adopted an α-helical conformation.

Our work revealed no significant differences between the random-coil and short helical hACE2 derivatives according to our bioreporter-based neutralization assay. However, in our 48 h pseudovirus entry inhibition assay, we clearly demonstrated the untapped potential of staple peptides. That situation was exemplified by compound **11** based on RCM cross link, which exhibited a six-fold increase in pseudovirus inhibition at 735 μM compared with its linear analogue. Therefore, even if our goal was to minimize the entropic cost associated with the adoption of the active conformation, additional data pertaining to the affinity of the ligand towards the SARS-CoV-2 Spike RBD and follow-on computational analyses hold promise for revealing the necessary kinetic perspective and elucidating how hydrocarbon peptide **11** efficiently inhibits pseudoviral entry.

To the best of our knowledge, this study is the first systematic investigation of staple screening performed on short peptide sequences. We have identified retention techniques for optimal formation of α-helical structures for further development of SARS-CoV-2 Spike RBD/hACE2 PPI inhibitors using this approach as a model system.

## Figures and Tables

**Figure 1 ijms-25-00166-f001:**
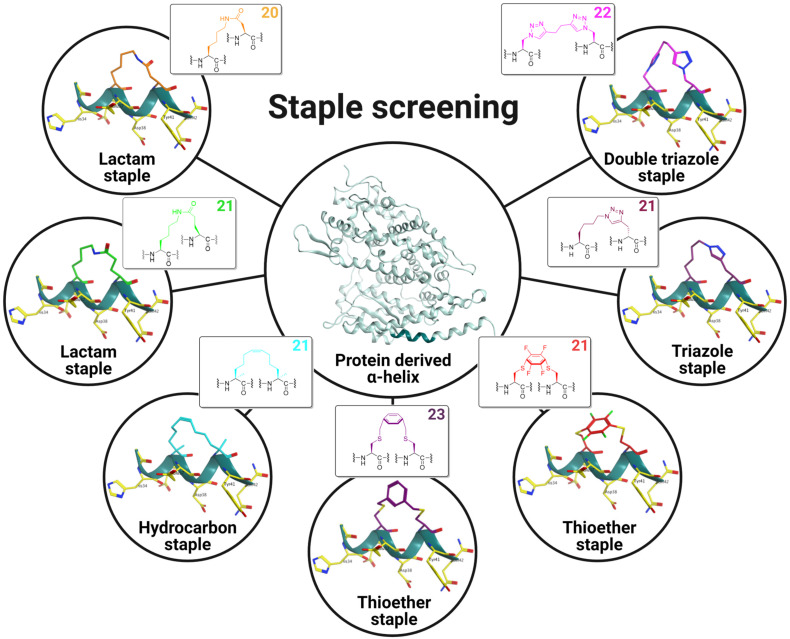
Staple screening performed on a short hACE2-derived α-helical sequence (His34-Gln42) (shown in pine green) via *i*, *i* + 4 side-to-side chain cyclizations by lactamization, olefin ring-closing metathesis (RCM), S-alkylation, S-arylation and copper(I)-catalyzed Huisgen 1,3-dipolar azide-alkyne cycloaddition (CuAAC). Ball-and-stick model staple representations are shown in the insets and macrocycle sizes are indicated in the upper-righthand corner. Residues from hACE2 involved in the SARS-CoV-2 S RBD/hACE2 interaction are shown as yellow sticks labeled with three-letter codes.

**Figure 2 ijms-25-00166-f002:**
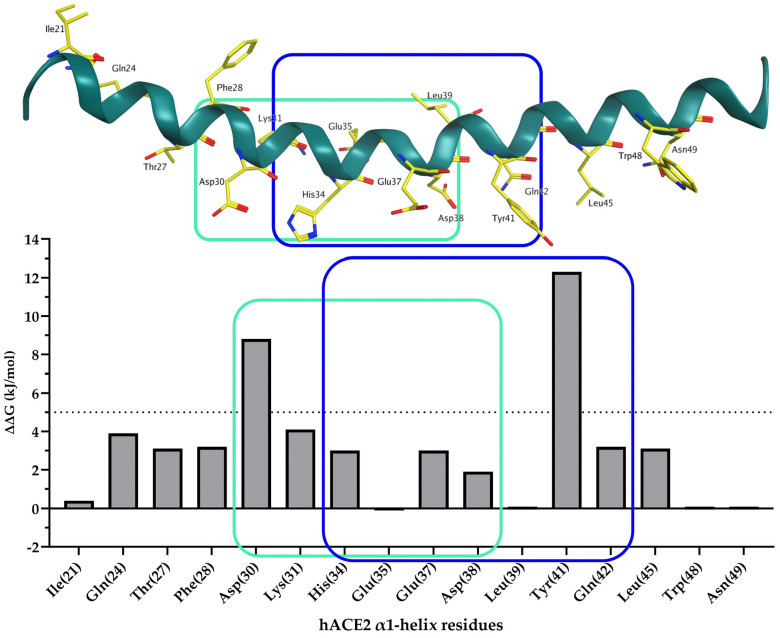
In silico alanine scan (BudeAlaScan ^a^) results for α1-helix (Ser19-Thr52) of hACE2 taken from the crystal structure of SARS-CoV-2 S RBD bound with hACE2 (PDB 6M0J). Residues from hACE2 involved in the SARS-CoV-2 S RBD/hACE2 interface are shown as yellow sticks labeled with three-letter codes. Selected hACE2-derived sequences (Asp30-Asp38) and (His34-Gln42) are outlined in light green and blue, respectively. ^a^ BudeAlaScan is an online software (version 1.0) available at https://pragmaticproteindesign.bio.ed.ac.uk/balas/ (accessed on 7 June 2023); it is only applicable to proteins consisting of natural amino acids.

**Figure 3 ijms-25-00166-f003:**
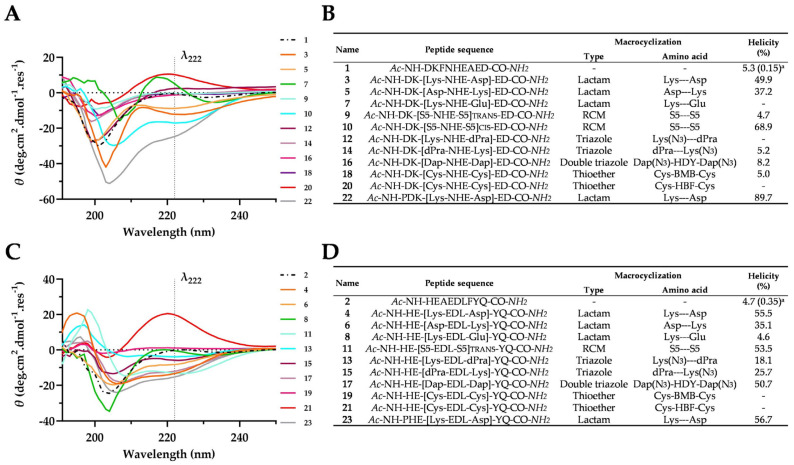
Data generated from Far-UV CD spectra for hACE2-derived peptides measured at a concentration of 100 µM in 10 mM sodium phosphate buffer at pH 7.4 and 25 °C. (**A**) Far-UV CD spectra of hACE2 (Asp30-Asp38)-derived linear and *i*, *i* + 4 staple peptides in molar ellipticity per residue. (**B**) Table listing the derivatives of **1** with a focus on the macrocyclization technique and measure-based calculated helicity (%) via CD. (**C**) CD spectra of hACE2 (His34-Gln42)-derived linear and *i*, *i* + 4 staple peptides in molar ellipticity per residue. (**D**) Table listing the derivatives of **2** with a focus on the macrocyclization technique and measure-based calculated helicity (%) via CD. HDY 1,5-Hexadiyne; BMB 1,4-Bis(bromomethyl)benzene; HFB hexafluorobenzene. ^a^ Predicted helicity (%) using AGADIR online software available at http://agadir.crg.es (accessed on 17 April 2023); it is only applicable to peptides consisting of natural amino acids.

**Figure 4 ijms-25-00166-f004:**
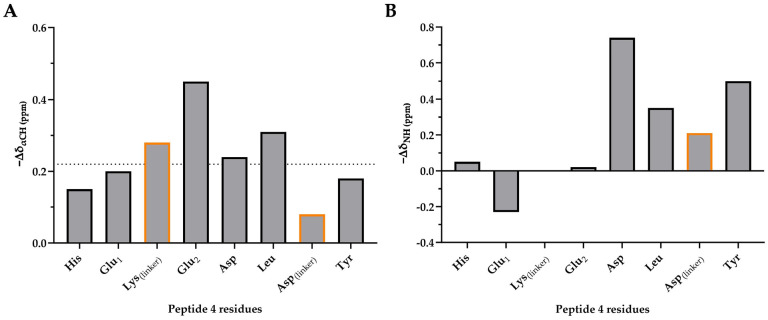
Secondary chemical shifts ^a^ of (**A**) CαH and (**B**) NH protons for peptide **4**. Secondary chemical shifts of linker-formed residues have been highlighted with an orange-colored outline. ^a^ Random-coil chemical shifts for 20 common amino acids followed by alanine were measured using a peptide with free N- and C-termini at pH 5.0 and 25 °C.

**Figure 5 ijms-25-00166-f005:**
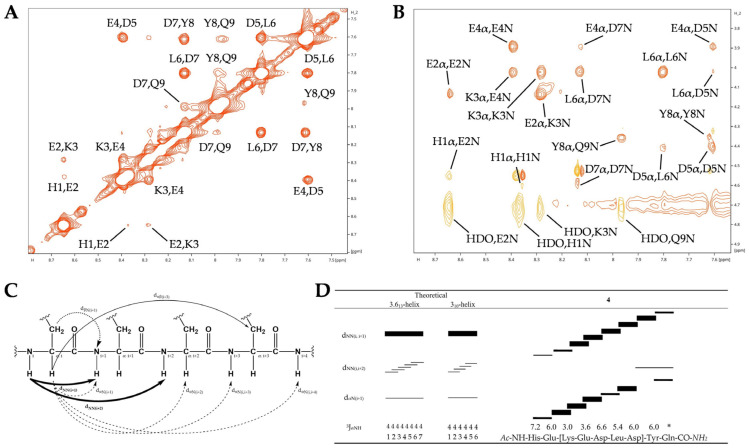
Characteristic short- and medium-range sequential NOEs for an α-helix assessment. (**A**) The d_NN_ and (**B**) d_αN_ regions of a NOESY spectrum recorded for peptide **4**. (**C**) Graphical illustration of sequential and medium-range ^1^H-^1^H distances in a peptide sequence. (**D**) Schematic representation of NOESY patterns involving NH and CαH protons observed in a NOESY spectrum recorded for α-helix compared with peptide **4**. The horizontal lines of various lengths indicate NOE connectivities between protons of peptide sequences; the thicknesses of the lines is proportional to the observed strong, medium, and weak NOEs signal intensities. * The protons could not be assigned unambiguously.

**Figure 6 ijms-25-00166-f006:**
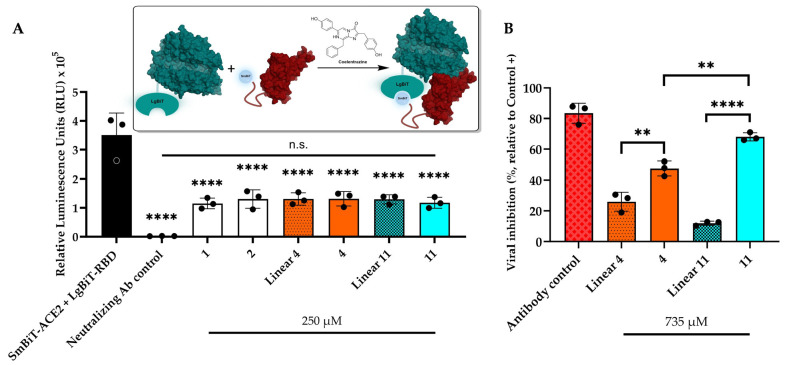
The SARS-CoV-2 Spike RBD/hACE2 inhibition assessment involving a bioluminescence-based bioreporter assay and a pseudovirus-based entry inhibition assay. (**A**) Split-luciferase bioreporter assay demonstrating the disruptor capacity of peptides **1**, **2**, **linear 4**, **4**, **linear 11** and **11**. Asterisks indicate a statistically significant difference between the RLUs measured for SmBiT-ACE2 + LgBiT-RBD and the neutralizing Ab control or an individual peptide. (**B**) Antiviral activity of peptides **linear 4**, **4**, **linear 11** and **11** was assessed using a SARS-CoV-2 pseudovirus carrying a fluorescent reporter gene and HEK-293T-hACE2 cells transfected with TMPRSS2. The fluorescence data (RFUs) were converted into percentages via normalization with the infection-free control (Dulbecco’s Eagle Medium; DMEM) set as 0%; we furthermore considered that the pseudovirus-only samples represented maximum infection (100%) using GraphPad Prism software (version 9.3.1). Both experiments were repeated in triplicate three times, and the data are expressed as means ± standard error of the mean (SEM) (error bars). The means of more than two groups were compared using one-way ANOVA with Tukey’s multiple comparison correction. For all analyses, **** *p* < 0.0001; ** 0.0017 ≤
*p*
≤ 0.0025; n.s., not significant.

**Figure 7 ijms-25-00166-f007:**
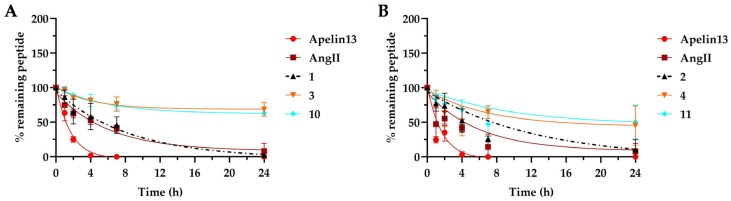
Plasma stability measured in rat plasma over 24 h of incubation at 37 °C. (**A**) The proteolytic stability of hACE2 (Asp30-Asp38)-derived peptides recorded as a function of degraded peptide over 24 h. (**B**) The proteolytic stability of hACE2 (His34-Gln42)-derived peptides recorded as a function of degraded peptide over 24 h. The data are plotted as means and SEMs of duplicate independent experiments. The percentage of residual peptide was monitored using UPLC-MS. All of the experiments were repeated three times.

**Table 1 ijms-25-00166-t001:** Proton chemical shift assignments for compound **4**.

Residue	NH	CαH	CβH	CγH	CδH	CεH
N-terminus						
His	8.37	4.58	3.03, 3.1	-	7.1, *	2, *
Glu_2_	8.65	4.15	1.87, 2.01	2.19, *	-	-
Lys_(linker)_	8.29	4.04	1.72, 1.80	1.30, 1.47	1.12, *	*, *
Glu_1_	8.40	3.90	1.92	2.21, *	-	-
Asp	7.60	4.4	2.58, 263	-	-	-
Leu	7.81	4.03	1.64, 1.76	1.71	0.74, 0.77	-
Asp_(linker)_	8.13	4.56	2.50, 2.72	-	-	-
Tyr	7.62	4.37	2.97, 3.01	-	*, *	*, *
Gln	*	*	*, *	*, *	-	*, *
C-terminus						

* The protons could not be assigned unambiguously.

## Data Availability

Data are contained within the article.

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
