# Peer review of "Comparative Analysis of Cyclization Techniques in Stapled Peptides: Structural Insights into Protein–Protein Interactions in a SARS-CoV-2 Spike RBD/hACE2 Model System"

_ijms, 2023, doi:10.3390/ijms25010166_

Round 1

Reviewer 1 Report

Comments and Suggestions for Authors

In this paper, the authors reported comparative study of several side chain stapling methods to introduce helix structure in short peptides. The authors synthesized several kinds of staple peptides and intensively evaluated their structures using CD and NMR. Eventually, RCM stapled peptide 11 applied biological studies. The results reported in the paper support the conclusion, but following minor discussion could improve the quality of the manuscript.

Minor comment:

Peptide 11 with trans olefin-containing side chain tether showed high helicity, and the authors discussed that trans isomer is suitable for helix formation in line 282. But, in case of peptide 9 and 10, which are also RCM peptides, cis isomer 10 showed helix structure. Please discuss this point more in the manuscript.

Author Response

Subject: Response to the Reviewer Comments on Manuscript ID ijms-2752679, titled "Comparative Analysis of Cyclization Techniques in Stapled Peptides: Structural Insights into PPI in the SARS-CoV-2 Spike RBD/hACE2 Model System" submitted to International Journal of Molecular Sciences.

Dear Ms. Rachel Zhou,

We would like to express our sincere gratitude to you and the reviewers for the time, commitment, and effort devoted to the evaluation of our manuscript.

First, we are grateful for the positive recognition of the overall quality of our research, particularly the meticulous understanding of our experimental design and the in vitro experiments conducted. We note that both reviewers recognized the diverse stapling techniques employed for the stabilization of the α-helical structure, based on the detailed analysis of hACE2 residues critical for binding to the RBD domain of SARS-CoV-2. Both reviewers also acknowledged structural analysis using CD, NMR, and in vitro assessment of antiviral activity in bioreporter- and pseudovirus-based inhibition assays.

Second, we appreciate the constructive comments provided to us, which we believe have helped to refine our manuscript with the aim of enhancing its clarity and bringing greater consistency to our findings. We would like to assure you that we have taken all comments seriously and are confident that we have addressed all of them to substantially improve the quality and robustness of our manuscript. Below, we briefly outline our responses to the major and minor concerns raised by the reviewers.

Third, recognizing the importance of adhering to the highest standards of scientific language, we have focused on revising the English language used in our manuscript with the goal of enhancing the comprehensibility and consistency of our research data.

Thank you again for considering our work and providing us with insightful feedback. We look forward to making any necessary changes and resubmitting our manuscript for your kind attention and that of the reviewers.

Best regards,

Pierre-Luc Boudreault

forward to making any necessary changes and resubmitting our manuscript for your kind attention and that of the reviewers.

Best regards,

Pierre-Luc Boudreault

To ease the reading, the comments of reviewers are highlighted in gray.

Text that has been added to the manuscript is highlighted in yellow.

The rectifications in the English language are visually represented by the use of green highlights.

Responses to the reviewer 1

Comment 1: Peptide 11 with trans olefin-containing side chain tether showed high helicity, and the authors discussed that trans isomer is suitable for helix formation in line 282. But, in case of peptide 9 and 10, which are also RCM peptides, cis isomer 10 showed helix structure. Please discuss this point more in the manuscript.

As stated by the reviewer, the TRANS isomer of the hydrocarbon stapled peptide 11 displays the typical a-helical Far-UV CD spectra (190-250 nm) with one maximum between 190 and 195 nm and two negative minima at approximately 208 and 222 nm, as previously reported [1]. This result has prompted us to suggest that the TRANS configuration of the olefin was suitable for the stabilization of helical structure. (line 299-301)

However, in the case of peptide 9 (TRANS) and 10 (CIS) and as pointed out by the reviewer it is the latter that shows more helical structure on the basis of mathematical evaluation of helicity at 222 nm. The CIS isomer of the hydrocarbon-stapled peptide 10 does not display the characteristic shape of the CD curve expected for an α-helical conformation and further extrapolation by BeStSel was performed for that reason. We point out that this degree of helicity (%) is due to a large population of an antiparallel right-twisted β-sheet conformation (line 238), as one of the features of the Far-UV CD spectra of β-sheets includes a negative band between 217 and 218 nm.

Hence, it appears that our original suggestion made for peptide 11 cannot be generalized. Thus, it is possible that the configuration of the olefin that favors a-helical structure is context or primary structure dependent. It is not unreasonable to assume that the i, i±3 interactions involving the bulky S5 residues are different for the two series. In this context, it is plausible that for the case of peptide 10, the CIS conformer is more amenable to stabilize an a-helical structure rather than the TRANS conformer as observed in peptide 11.

Since we appreciate the reviewer's insights, we have incorporated additional information into the manuscript.

“The Far-UV CD spectra recorded for the hACE2-derived staple peptides (Asp30-Asp38) did not display the characteristic shape expected for an α-helical conformation (i.e., one maximum between 190 and 195 nm and two negative minima at approximately 208 and 222 nm, as previously reported [1]).” (page 8, line 232-235)

Therefore, the high percentage of calculated helicity in compounds 10 and 22 can be explained on the basis of the contribution of a strong, negative absorbance band within the 217–218 nm spectral range of the Far-UV CD spectra. Such features are typical of a right-twisted β-sheet [1,2].” (page 8, line 242-246)

Compound 11 attained a helical content of up to 53.5% based on CD. This finding suggests clear superiority of the TRANS isomer in terms of helicity but only due to the favorable entropic conformational flexibility of the double bonds in the RCM reaction context.” (page 10, line 298-301)

Reviewer 2 Report

Comments and Suggestions for Authors

The manuscript entitled “Comparative Analysis of Cyclization Techniques in Stapled 2 Peptides: Structural Insights into PPI in the SARS-CoV-2 Spike 3 RBD/hACE2 Model System” by Ferková et al. is focused on the design of small peptidomimetics able to inhibit the SARS-CoV-2 Spike RBD interaction with the α1-helix of hACE2. Specifically, the design strategy was undertaken using the stapled technique, which is a key methodology for stabilising peptides in an α-helical structure. The authors have first collected four PDB models of SARS-CoV-2 RBD models interacting with hACE2 and they have analyzed, with the in-silico alanine scan, the hACE2 residues mostly critical for the binding with the RBD domain. This analysis led to select two short hACE2 native sequences to be stapled: peptide 1 (Asp30-Asp38) and peptide 2 (His34-Gln42). These two sequences were then covalently bound to various staple chemical types: lactam, hydrocarbon, triazole and double triazole and thioether. The designed stapled peptides were then synthesized with the SPPS technique and were tested for their capability to maintain the α-helical content (Far-UV CD, helicity and NMR), stereochemical relationships with NOESY and antiviral tests via bioluminescence- and pseudovirus-based entry inhibition assays. Finally, resistance to proteolytic degradation was measured by UPLC-MS, revealing that the most promising staple peptides 4 and 11 were significantly more stable in rat plasma with respect to the natural linear sequences.

Globally the work was well-conducted and it was enriched with both biophysical and in vitro experiments that support the aim of the study. Despite this, I believe that it needs of some major revisions:

Major points: 

1. The work was undertaken with a conventional structure-guided stapling screening strategy. The peptide design was performed, in fact, through the inspection of only a limited number among the available SARS-CoV-2 RBD/hACE2 complexes. Using only four X-Ray structures could be, in fact, not statistically significant to identify the critical residues to the RBD/hACE2 recognition. Furthermore, X-Ray structures represent only one snapshot of the possible interaction patterns between RBD and hACE2 interfaces. A possible solution to this issue, could be the use of an ensemble of conformations (i.e collected from Molecular Dynamics simulations), as well as the use of a greater number of experimental structures to compare the “intrinsic similarity”. I think that this limiting aspect of the work should be highlighted along text and especially in the “Conclusions” section.

2 2.  Page 5, lines 145-149: It is not clear to me the rationale behind the selection of the four PDB models. Authors should better explain the reason of such models.

33. The Receptor binding motif (RBM) of the Spike-RBD domain, is not conserved in other SARS-CoV-2 variants. For instance, the Omicron variant has a significantly larger number of mutations in the RBM motif and one of them concerns Gly496 in the BA.1 subvariant. Despite several PDB RBD/hACE2 Omicron variants are available in the PDB, it seems that the authors have not discussed or compared the effects of the RBM mutations in the design of the PPI stapled peptides. This could lead to a variant-specific peptide design strategy and this aspect must be highlighted and analyzed in the text.

Minor point:

SSome references of the Figure in the Supporting Information are missing in the text

Comments on the Quality of English Language

    English revision is required.

Author Response

Subject: Response to the Reviewer Comments on Manuscript ID ijms-2752679, titled "Comparative Analysis of Cyclization Techniques in Stapled Peptides: Structural Insights into PPI in the SARS-CoV-2 Spike RBD/hACE2 Model System" submitted to International Journal of Molecular Sciences.

Dear Ms. Rachel Zhou,

We would like to express our sincere gratitude to you and the reviewers for the time, commitment, and effort devoted to the evaluation of our manuscript.

First, we are grateful for the positive recognition of the overall quality of our research, particularly the meticulous understanding of our experimental design and the in vitro experiments conducted. We note that both reviewers recognized the diverse stapling techniques employed for the stabilization of the α-helical structure, based on the detailed analysis of hACE2 residues critical for binding to the RBD domain of SARS-CoV-2. Both reviewers also acknowledged structural analysis using CD, NMR, and in vitro assessment of antiviral activity in bioreporter- and pseudovirus-based inhibition assays.

Second, we appreciate the constructive comments provided to us, which we believe have helped to refine our manuscript with the aim of enhancing its clarity and bringing greater consistency to our findings. We would like to assure you that we have taken all comments seriously and are confident that we have addressed all of them to substantially improve the quality and robustness of our manuscript. Below, we briefly outline our responses to the major and minor concerns raised by the reviewers.

Third, recognizing the importance of adhering to the highest standards of scientific language, we have focused on revising the English language used in our manuscript with the goal of enhancing the comprehensibility and consistency of our research data.

Thank you again for considering our work and providing us with insightful feedback. We look forward to making any necessary changes and resubmitting our manuscript for your kind attention and that of the reviewers.

Best regards,

Pierre-Luc Boudreault

To ease the reading, the comments of reviewers are highlighted in gray.

Text that has been added to the manuscript is highlighted in yellow.

The rectifications in the English language are visually represented by the use of green highlights.

Responses to the reviewer 2

Comment 1: The work was undertaken with a conventional structure-guided stapling screening strategy. The peptide design was performed, in fact, through the inspection of only a limited number among the available SARS-CoV-2 RBD/hACE2 complexes. Using only four X-Ray structures could be, in fact, not statistically significant to identify the critical residues to the RBD/hACE2 recognition. Furthermore, X-Ray structures represent only one snapshot of the possible interaction patterns between RBD and hACE2 interfaces. A possible solution to this issue, could be the use of an ensemble of conformations (i.e collected from Molecular Dynamics simulations), as well as the use of a greater number of experimental structures to compare the “intrinsic similarity”. I think that this limiting aspect of the work should be highlighted along text and especially in the “Conclusions” section.

We agree and acknowledge the reviewer’s point that the use of only four X-Ray structures may, in fact, not be statistically significant in identifying residues critical for RBD/hACE2 recognition. In fact, we used only four PDB files to point out that numerous residues are involved in the PPI between the RBD of SARS-CoV-2 and hACE2 and that changes in the sharing of a different binding partner already exist in four structures resolved by X-ray or Cryo-EM (line 153-154). Furthermore, in silico alanine scan (BudeAlaScan) was performed on protein-derived sequence from PDB file 6M0J (line 166-168), as this crystal structure is currently selected by the scientific community and has even been highlighted for its accuracy [3].

We believe that the main text meets the journal's standards, but if the editor suggests that it would be better to improve it, we can do so.

“These subtle differences that we observed when we analyzed the four resolved structures suggest that the SARS-CoV-2 RBD/hACE2 interface is highly complementary and dynamic in nature.” (page 5, line 154-156)

Using an in silico alanine scan (BudeAlaScan) performed on the α1-helix of hACE2 taken from the crystal structure of SARS-CoV-2 S RBD bound with ACE2 (PDB 6M0J), we identified residues critical to binding.” (page 5, line 166-168)

Figure 2. In silico alanine scan (BudeAlaScana) results for α1-helix (Ser19-Thr52) of hACE2 taken from the crystal structure of SARS-CoV-2 S RBD bound with hACE2 (PDB 6M0J).” (page 6, line 177-178)

Comment 2 : Page 5, lines 145-149: It is not clear to me the rationale behind the selection of the four PDB models. Authors should better explain the reason of such models.

The selection of PDB files, namely 6M0J, 6VW1, 6LZG, and 6M17, was based on their accessibility at the time the project was launched. Additionally, our rationale includes the deliberate incorporation of diversity involving the technique employed, the resolution, and the specific target proteins investigated. This encompasses a range of considerations, such as examining SARS-CoV-2 chimeric structures, as well as taking into account the presence or absence of antibodies in the systems studied.

These early and highly accurate [3] structural insights provided a blueprint for our rational structure-based design of peptidomimetics.” (page 5, line 145-146)

The models were selected on the basis of their diversity of techniques, their resolution and the specific target proteins they investigated.” (page 5, line 149-150)

“4.3.7 Peptide design and visualization

The rational design of the peptides was based on previously published high-resolution structures: PDB 6M0J [4] (Crystal structure of SARS-CoV-2 S RBD bound with ACE2, X-ray diffraction, 2.45 Å, 2020-03-18), PDB 6VW1 [5] (Structure of SARS-CoV-2 chimeric RBD complexed with its receptor hACE2, X-ray, 2.68 Å, 2020-03-04), PDB 6LZG [6] (Structure of novel coronavirus S RBD complexed with its receptor ACE2, X-ray diffraction, 2.50 Å, 2020-03-18 ) and PDB 6M17 [7] (The 2019-nCoV RBD/ACE2-B0AT1 complex, electron microscopy, 2.90 Å, 2020-03-11).” (page 22, line 750-757)

Comment 3 : The Receptor binding motif (RBM) of the Spike-RBD domain, is not conserved in other SARS-CoV-2 variants. For instance, the Omicron variant has a significantly larger number of mutations in the RBM motif and one of them concerns Gly496 in the BA.1 subvariant. Despite several PDB RBD/hACE2 Omicron variants are available in the PDB, it seems that the authors have not discussed or compared the effects of the RBM mutations in the design of the PPI stapled peptides. This could lead to a variant-specific peptide design strategy and this aspect must be highlighted and analyzed in the text.

We acknowledge the reviewer’s suggestion to enhance the main text by mentioning the consequence of the SARS-CoV-2 mutations present in the different variants. However, the design of our peptide is not based on the SARS-CoV-2 sequence, but is derived from the hACE2 protein.

Variants of human ACE2 in different populations could affect the binding affinity between SARS-CoV-2 and hACE2, as previously reported [8]. Nevertheless, we feel that this part of the study should be the subject of another publication, as all the mutations mentioned above - hACE2 (G211R, D206G, K341R, R219C, I468V) are considered rare (<1%) and would require large-scale sequencing projects performed beforehand. It should be noted that none of these variants includes a mutation at the SARS-CoV-2/hACE2 interface and would therefore have no impact on the rational design of our peptides. In addition, a previous study shows that no significant changes were observed in the overall structural conformation of ACE2 protein variants and substituted amino acids found in the SARS-CoV-2/ACE2 contact area [9].

Nonetheless, mutations in SARS-CoV-2, specifically N501Y, K417N, E484K, and N439K, have been shown to enhance interaction with the hACE2 target [10]. We hypothesize that if these mutations result in increased binding affinity to hACE2, a similar trend could be observed with our compounds. This additional advantage supports the endogenous peptide-based design.

As we very much appreciate the reviewers’ opinion, we added variants of hACE2 related information to the manuscript for greater precision.

We note that none of the previously described rare (<1%) hACE2 variants included a mutation (K26R, D206G G211R, R219C, K341R, I468V) in the studied contact area of SARS-CoV-2/hACE2 [8], which would have impacted our rational peptide design.” (page 5 , line 162-165)

Comment 4 : Some references of the Figure in the Supporting Information are missing in the text.

A comprehensive inquiry was conducted to address the absence of Supporting Information Figure specifications. We believe that Figure S1-S19 have been properly referenced in the main text of the manuscript. Furthermore, a meticulous examination was undertaken to locate previously unaccounted Supplementary Information corresponding to the Table specifications. Consequently, the main text of the manuscript “(Table 1 and S5-S7, Figure S7-S9 and Figure S10-S13)”  has been revised for enhanced clarity as follows:

“(Figure S7S13, Table 1 and Table S5S7).” (page 9, line 272-273)

And

We isolated peptides with a purity exceeding 95% for use in subsequent experiments. (Table S9–S12) ” (page 18, line 591-593)

Suitability for the Journal: While we respect the reviewer’s opinion regarding the relevance of our paper submitted to Special Issue Advances in Protein-Protein Interactions 2.0, we believe that our research aligns with the focus of the International Journal of Molecular Sciences, given the emphasis on the design, structural studies and development of novel PPI inhibitors. We hope that the revisions will clarify the relevance and contribution of our work to the field of medicinal chemistry, making it favorably comparable to other articles related to SARS-CoV-2 S RBD/hACE2 PPI studies.

  1. Bakshi, K.; Liyanage, M.R.; Volkin, D.B.; Middaugh, C.R. Circular Dichroism of Peptides. Methods in Molecular Biology 2014, 1088, 247–253, doi:10.1007/978-1-62703-673-3_17.
  2. Micsonai, A.; Wien, F.; Kernya, L.; Lee, Y.H.; Goto, Y.; Réfrégiers, M.; Kardos, J. Accurate Secondary Structure Prediction and Fold Recognition for Circular Dichroism Spectroscopy. Proc Natl Acad Sci U S A 2015, 112, E3095–E3103, doi:10.1073/pnas.1500851112.
  3. Xiaoqiang Huang, R.P., Y.Z. De Novo Design of Protein Peptides to Block Association of the SARSCoV-2 Spike Protein with Human ACE2. Aging (Albany NY) 2020, 12.
  4. Lan, J.; Ge, J.; Yu, J.; Shan, S.; Zhou, H.; Fan, S.; Zhang, Q.; Shi, X.; Wang, Q.; Zhang, L.; et al. Structure of the SARS-CoV-2 Spike Receptor-Binding Domain Bound to the ACE2 Receptor. Nature 2020, 581, 215–220, doi:10.1038/s41586-020-2180-5.
  5. Shang, J.; Ye, G.; Shi, K.; Wan, Y.; Luo, C.; Aihara, H.; Geng, Q.; Auerbach, A.; Li, F. Structural Basis of Receptor Recognition by SARS-CoV-2. Nature 2020, 581, 221–224, doi:10.1038/s41586-020-2179-y.
  6. Wang, Q.; Zhang, Y.; Wu, L.; Niu, S.; Song, C.; Zhang, Z.; Lu, G.; Qiao, C.; Hu, Y.; Yuen, K.Y.; et al. Structural and Functional Basis of SARS-CoV-2 Entry by Using Human ACE2. Cell 2020, 181, 894-904.e9, doi:10.1016/j.cell.2020.03.045.
  7. Yan, R.; Zhang, Y.; Li, Y.; Xia, L.; Guo, Y.; Zhou, Q. Structural Basis for the Recognition of SARS-CoV-2 by Full-Length Human ACE2; 2020; Vol. 367;.
  8. Ali, F.; Elserafy, M.; Alkordi, M.H.; Amin, M. ACE2 Coding Variants in Different Populations and Their Potential Impact on SARS-CoV-2 Binding Affinity. Biochem Biophys Rep 2020, 24, doi:10.1016/j.bbrep.2020.100798.
  9. Hussain, M.; Jabeen, N.; Raza, F.; Shabbir, S.; Baig, A.A.; Amanullah, A.; Aziz, B. Structural Variations in Human ACE2 May Influence Its Binding with SARS-CoV-2 Spike Protein. J Med Virol 2020, 92, 1580–1586, doi:10.1002/jmv.25832.
  10. Han, Y.; Wang, Z.; Wei, Z.; Schapiro, I.; Li, J. Binding Affinity and Mechanisms of SARS-CoV-2 Variants. Comput Struct Biotechnol J 2021, 19, 4184–4191, doi:10.1016/j.csbj.2021.07.026.

Round 2

Reviewer 2 Report

Comments and Suggestions for Authors

Authors have adequately addressed the major points required